**Data Availability Statement:** De-identified study data are available at OSF: https://osf.io/je238/.

# Planning and commitment prompts to encourage reporting of HIV self-test results: A cluster randomized pragmatic trial in Tshwane District, South Africa

Alison M. Buttenheim[1]*, Laura Schmucker[2], Noora Marcus[2], Mothepane Phatsoane[3], Vanessa Msolomba[3], Naleni Rhagnath[3], Mohammed Majam[3], François Venter[3], Harsha Thirumurthy[2]

1 Department of Family and Community Health, School of Nursing, University of Pennsylvania, Philadelphia, Pennsylvania, United States of America, 2 Department of Medical Ethics and Health Policy, Perelman School of Medicine, University of Pennsylvania, Philadelphia, Pennsylvania, United States of America, 3 Ezintsha, a Sub-Division of Wits Reproductive Health and HIV Institute, University of Witwatersrand, Johannesburg, Gauteng, South Africa

* abutt@upenn.edu

## Abstract

Reporting of HIV self-test results to encourage linkage to HIV care for those who receive a positive test result is a common challenge faced by HIV self-testing programs. The impact of self-testing programs is diminished if individuals who obtain a self-test do not use the test or seek confirmatory testing and initiate HIV treatment following a positive result. We conducted a cluster randomized trial of two interventions designed to increase reporting of HIV self-test results: a "plan and commit" intervention that leveraged insights from behavioral economics, and an enhanced usual care version of the standard HIV self-test community distribution protocol that promoted the importance of reporting results. The trial was conducted at community distribution sites for HIV self-tests in Tshwane Metropolitan Municipality, Gauteng Province, South Africa. The primary outcome was reporting of self-test results via a WhatsApp messaging system. We recruited 1,478 participants at 13 distribution sites over 24 days. In the plan and commit condition, 63/731 participants (8.7%) reported their test results via WhatsApp, compared to 59/747 participants (7.9%) in the enhanced usual care condition (n.s., $p = 0.61$). During the study period, 101/3,199 individuals (3.1%) who received a self-test under the standard protocol reported test results via WhatsApp, a significant difference across the three arms ($p < .00001$). Our results suggest that boosting the reporting of self-test results can be done solely through increasing the salience of the importance of reporting and a clear explanation of the procedure for reporting results.

**Trial Registration:** ClinicalTrials.gov: NCT03898557.

## Introduction

Achieving the UNAIDS 95-95-95 targets is essential for ending the global HIV epidemic [1]. Despite progress towards these targets in sub-Saharan Africa, promoting HIV testing among

**Funding:** This study was funded by the Bill & Melinda Gates Foundation (gatesfoundation.org) through award INV-008318 (HT). The funders had no role in study design, data collection and analysis, decision to publish, or preparation of the manuscript.

**Competing interests:** The authors have declared that no competing interests exist.

people living with HIV (PLHIV) and linking them to HIV care has proven challenging [2]. To increase testing coverage among hard-to-reach individuals at high risk of HIV infection, HIV self-testing (HIVST) has been widely used in recent years. Private and convenient, HIVST offers advantages over other testing modalities and can overcome barriers to testing for priority populations who may be less likely to seek facility-based approaches [3,4]. The World Health Organization's HIV testing guidelines recommend the scale-up of HIVST. In southern Africa, the STAR (Self-Test Africa) initiative, supported by UNITAID, has distributed over 700,000 self-tests to date.

Despite scale-up of this promising approach, behavioral barriers to self-test use and post-test results reporting linkage to care limit the effectiveness of HIVST [5]. The impact of HIVST programs is diminished if individuals pick up self-tests at distribution sites, but do not subsequently use the test or seek confirmatory testing following a positive self-test result. While picking up a self-test at a community distribution site suggests an intention to use the test and seek appropriate health care, following up on those intentions may be challenging due to several behavioral factors. These include procrastination stemming from present-biased decision-making, competing priorities, hassle factors, fear of obtaining a positive test result, and stigma associated with seeking care. Similar to many other screening and prevention behaviors, there can be a large gap between intention and action in HIVST use and linkage decisions.

Behavioral economics research suggests several low-cost interventions that can encourage individuals to follow through on their intentions to engage in specific health behaviors. Planning prompts, which prompt individuals to make a plan for undertaking a behavior, have shown promise in closing the intention-action gap for diverse actions including voting [6], influenza vaccination [7] and HIV testing [8]. Planning prompts encourage behavioral rehearsal and facilitate the formation of specific implementation intentions [9–12]. Additionally, commitment devices, even when the commitment is private and unenforceable, leverage people's intrinsic desire to maintain a consistent self-image and follow what they perceive to be the descriptive and injunctive norm [13]. Planning prompts and commitment devices are effective strategies for translating behavioral intentions into actions, particularly when barriers to the desired behavior include procrastination, prospective memory failures, and hassle factors. In this pragmatic randomized trial, we assessed the effectiveness of a simple planning prompt and commitment device for increasing reporting of HIV self-test results by individuals who obtained self-tests at community distribution sites in South Africa.

## Materials and methods

### Ethics statement

The STAR program was approved by the University of Witwatersrand. The study was approved by the University of Pennsylvania Institutional Review Board. A waiver of informed consent was granted under the umbrella STAR protocol. The trial was registered at clinical-trials.gov (NCT03898557).

Design and setting. We conducted a cluster randomized trial of a planning and commitment prompt as part of the STAR-supported HIVST program in South Africa run by the Ezintsha unit of Wits RHI at the University of Witwatersrand. Ezintsha established a staffed WhatsApp mobile messaging line to enable self-testers to upload screening test results (including a photo of their completed test strip) and receive information about next steps: confirmatory testing and linkage to care if positive, or advice about repeat testing if negative. However, usage of the WhatsApp line was initially low, with reporting rates below 5% in the early months of the program.

This study was conducted in 13 urban locations within Tshwane Metropolitan Municipality of Gauteng Province where Ezintsha's STAR program operated. STAR distributed oral fluid-based HIV tests (Oraquick Advance, Orasure Technologies, Bethlehem, PA, USA) from market centers and other densely populated areas. Individuals passing by community fixed-point distribution sites, such as hotspots including taxi ranks and shopping areas, were offered information about HIVST and could obtain a self-test for personal use after receiving instructions and a demonstration of how to use the test.

### Recruitment and testing

Community members who approached the STAR self-test distribution sites were eligible to participate in the study. Per the existing STAR protocol, inclusion criteria were that community members had to be ≥19 years of age and exclusion criteria was that they must not visibly intoxicated to receive a self-test. For each individual who accepted a test, program staff completed a brief form collecting socio-demographic information and prior HIV testing history. Individuals who consented to follow-up by the STAR program were asked to provide their mobile phone number. Unique barcode stickers were placed on the self-test and on the form that linked the self-test to a specific individual. Individuals obtaining a self-test could either take the test with them for later use or complete the test immediately in a private portable tent. As mandated by South Africa's National Department of Health, those who chose to take the self-test with them for later use were given a card with instructions for seeking confirmatory testing in the event of a positive self-test result. In the Ezintsha STAR program, this card also included the WhatsApp reporting number.

### Interventions

We compared a plan and commit prompt (PC) to an enhanced usual care (EUC) version of the standard STAR distribution instructions, which were printed on cards and given to individuals in their self-test packets. The EUC card explained how to report results via WhatsApp and emphasized the importance of reporting. The counselors' verbal script similarly emphasized the ease and confidentiality of WhatsApp reporting, and highlighted three reasons for reporting: 1. To ensure that the test was used correctly; 2. To get help interpreting the results; and 3. To get support and advice for linking to additional services and care if needed.

The PC intervention differed from EUC in two ways: First, the printed card placed in the testing packet had additional spaces on the reverse side to make a plan to take the test (e.g., location and timing of completing the test) and to commit to use the test by signing a brief statement. Second, the accompanying PC script pointed out the plan and commit features of the PC card and suggested how to use it. Importantly, the PC intervention did not require participants to complete the card at the time of test distribution nor did it require them to return the card to the testing staff. Completion of the card was entirely voluntary and was unobserved by the STAR program staff and the study team.

### Target population & power calculations

Our target sample size of 1,320 self-test recipients (660 per arm) gave us power to detect a 5-percentage point difference in WhatsApp reporting rates between the PC and EUC interventions, assuming a baseline reporting rate of 2.5% (as observed in previous iterations of the WhatsApp test result reporting initiative) and taking into account the clustered design with team-site-day as the unit of randomization. Each cluster of team-site-day pairs distributed self-tests until daily targets were met.

### Randomization

A cluster randomized design was used and the unit of randomization was the team-site-day. A team consisted of two STAR program staff working together to distribute tests, demonstrate their use, and complete the data collection form. Two study teams worked simultaneously at a selected testing site, in addition to other field staff distributing tests according to the standard protocol. Cluster randomization was used rather than individual randomization to ensure feasibility of intervention delivery and to align most closely with the existing test distribution programme. Each team pair distributed only EUC or PC cards in a given day.

Using a computer-generated randomization scheme, the first author (AB) assigned each team-site-day to the EUC or PC intervention. After the trained study team distributed 30 tests according to the assigned EUC or PC condition in a given day at a given test site, the team reverted to the standard protocol for test distribution for the remainder of the day. Participants and the study team were not blinded as to condition.

### Outcomes

The primary outcome was test results reported to the program's WhatsApp numbers as a proportion of tests distributed. The secondary outcome was the proportion of HIV-positive test results reported as proportion of tests distributed (i.e., reported yield). Outcomes were measured by STAR program nurses who staffed the separate WhatsApp numbers for each of the two interventions. Nurses were not blinded as to condition of the WhatsApp number they were staffing, but were also not directly involved in the design of the study and were not familiar with the distinctions between arms.

### Analysis

Following our pre-registered analysis plan, we compared test result reporting rates in the EUC and PC using chi-squared tests. As an exploratory analysis, we also compared the PC and EUC to the standard STAR protocol for test distributions during trial window using routine program data collected by the STAR program. Limitations in the recording of the WhatsApp results reporting meant that we were not able to assign individual results to clusters. In addition, the routine program data do not follow a similar clustering. therefore, the confidence intervals and the chi-squared analyses are not adjusted for clustering. See the study protocol (S1 Text) for more details.

## Results

Between May 28 and June 18, 2019, 1,478 participants were recruited and accepted a self-test at 13 distribution sites (see Fig 1 for study flow diagram). The distribution of gender, age and testing history of the study participants is shown in Table 1 (left-hand columns). The sample was about half female, with the majority of participants aged 18–24. About one third of participants had either never been tested for HIV or had not tested in the past year.

In the PC condition, 63/731 participants (8.7%) reporter their test results via WhatsApp, compared to 59/747 participants (7.9%) in the EUC (n.s., $p = 0.61$) (see Fig 1). While the study was not powered to detect subgroup differences in response to the interventions, an exploratory analysis by gender suggests that Plan & Commit was more effective then Enhanced Usual Control among women (35/365, 9.5%, vs. 26/398, 6.5%), while the opposite was true for men (28/361, 7.8%, vs. 33/348, 9.5%).

During the study period, 101/3199 individuals (3.1%) who received a self-test under the standard protocol reported test results via WhatsApp, a significant difference across the three

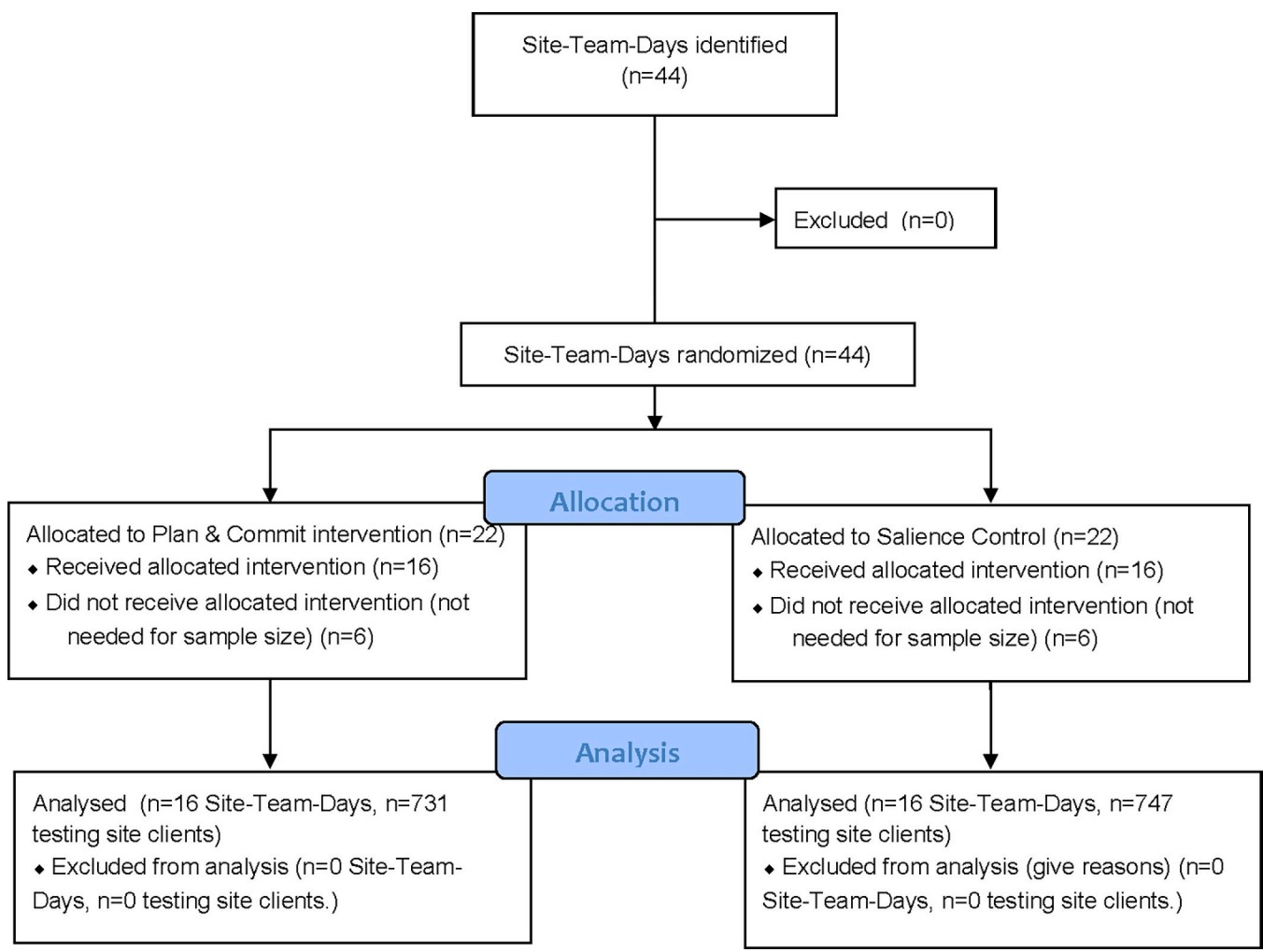

**Fig 1. Study flow diagram.**

arms $(p < .00001)$. No HIV-positive test results were reported PC and EUC intervention groups. Among the 3,199 other self-tests distributed under the standard protocol during the interventions period, no positive tests were reported (see Fig 2).

## Discussion

In this pragmatic randomized field trial, we tested two interventions hypothesized to increase the likelihood that individuals receiving HIV self-tests would report their test results back to the testing program. An enhanced usual care intervention emphasized the importance of reporting results and explained the process for doing so, while a "plan and commit" intervention added a planning prompt and a voluntary commitment pledge to use the test and report self-test results. Both interventions had similar test result reporting rates that were more than double the reporting rates observed in the HIVST program's standard protocol.

Our results suggest that boosting the reporting of self-test results can be done solely through increasing the salience of results reporting by emphasizing why reporting is important and by providing a clear explanation of the procedure for reporting results. The plan and commit intervention as implemented in this study did not significantly increase test result reporting

**Table 1. Characteristics of recipients of HIV self-screening tests (HIVSST) and those reporting test results via WhatsApp messaging, Tshwane Metropolitan Municipal District, 2019.**

| | HIVSST Distribution | | HIVSST Test Result Reports | |
|---|---|---|---|---|
| | Plan & Commit (N = 731) | Enhanced Usual Care (N = 747) | Plan & Commit (N = 63) | Enhanced Usual Care (N = 59) |
| Gender | | | | |
| Female | 365 (49.9%) | 398 (53.3%) | 35 (55.6%) | 26 (44.1%) |
| Male | 361 (49.4%) | 348 (46.6%) | 28 (44.4%) | 33 (56.0%) |
| Trans or Not Reported | 5 (0.7%) | 1 (0.1%) | 0 (0%) | 0 (0.0%) |
| Age | | | | |
| 18–24 | 390 (53.4%) | 395 (52.9%) | 36 (57.1%) | 28 (47.5%) |
| 25–39 | 280 (38.3%) | 305 (40.8%) | 27 (42.9%) | 25 (42.4%) |
| 40+ | 57 (7.8%) | 46 (6.2%) | 0 (0%) | 6 (10.2%) |
| Not reported | 4 (0.6%) | 1 (0.1%) | 0 (0%) | 0 (0%) |
| Most recent HIV test | | | | |
| Never tested | 49 (6.7%) | 82 (11.0%) | | |
| 0–3 months | 81 (11.1%) | 88 (11.8%) | | |
| 3–12 months | 368 (50.4%) | 372 (49.8%) | | |
| More than 12 months | 225 (30.8%) | 197 (26.4%) | | |
| Not reported | 8 (1.1%) | 8 (1.07%) | | |

above and beyond enhanced usual care. One conclusion from this study is that the mechanisms through which planning prompts and commitment devices work may be similar to those of a salience intervention: these strategies all drawing people's attention to the importance of the behavior and prompt them to complete the behavior at a future time. Our results

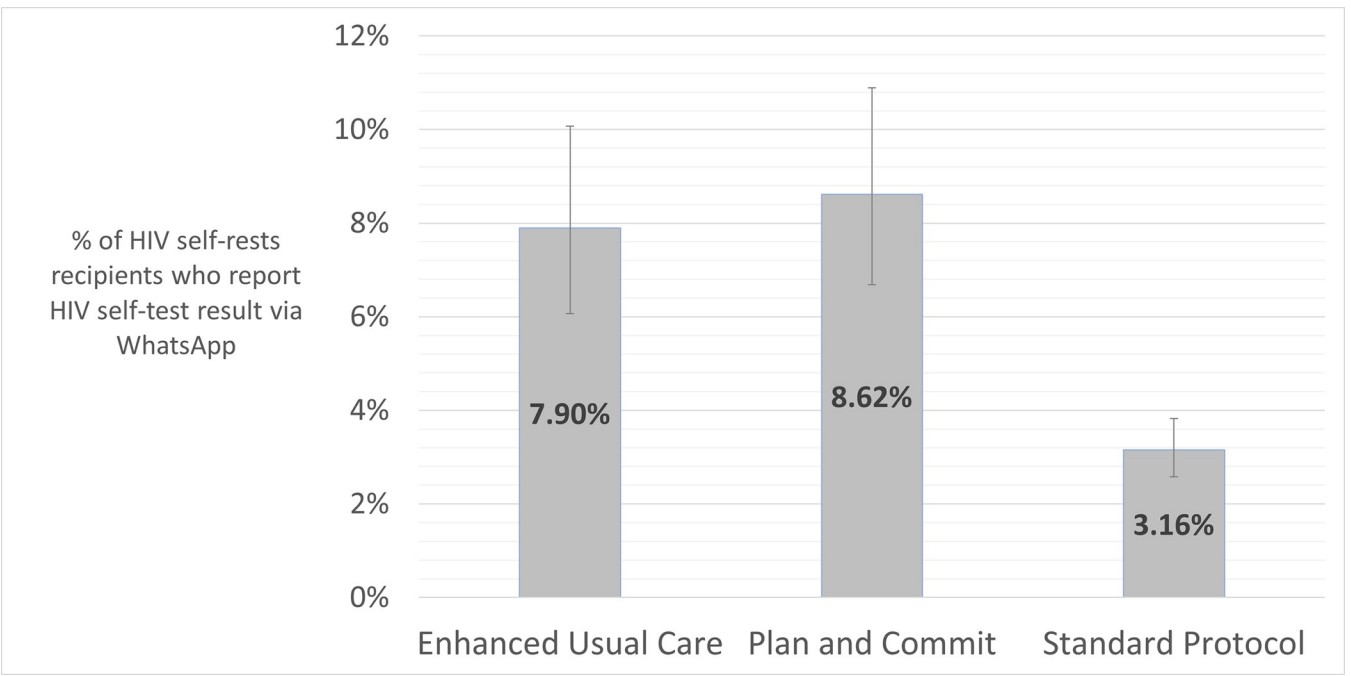

**Fig 2. Proportion of participants reporting HIV self-screening test results via WhatsApp messaging, by treatment arm, Tshwane Metropolitan Municipal District, 2019 (N = 4,677 total participants).**

are consistent with previous studies showing no changes in behavior due to planning prompts [14–17], and a recent trial that showed planning prompts resulted in small but statistically insignificant changes in HIV testing uptake by men in Uganda [8]. However, they are in contrast with other studies demonstrating the effectiveness of planning prompts over and above reminders, information provision, or education [18].

Our study has some important limitations. To maximize feasibility and generalizability, participants were not asked to complete the planning and commitment prompts at the time of self-test distribution. Completing the prompts at the distribution site might result in higher test result reporting rates. We were also unable to assess whether participants who did not report test results had used the self-test but not reported results, or if they did not use the self-tests at all. Some challenges with linking WhatsApp reporting results to test distribution also meant that we were unable to evaluate additional outcomes of interest including time to test result reporting, and prior testing history as a predictor of results reporting or as a treatment modifier. Finally, our decision to distribute tests following the Plan & Commit and Enhanced Usual Care protocols at the beginning of the testing day, followed by the standard protocol once the requisite number of intervention tests had been distributed, could potentially introduce a biased comparison of the two interventions to the standard protocol if community members stopping by the test distribution site at the beginning of the day were systematically different from those arriving later in the day. However, this would only affect the results of our exploratory analysis, and the risk of bias is attenuated by the fact that many standard protocol tests were distributed at sites that were not an intervention site in any given day and were therefore distributed throughout the full testing day.

A strength of the study is the light-touch, very low-cost interventions that are both feasible to incorporate into HIVST programs and easy to evaluate, with only 13 days in the field needed to accrue the desired sample size. This underscores the potential to rapidly design and evaluate similar low-cost, light-touch "nudges" that can contribute to progress on reaching the UNAIDS 2025 targets.

HIVST has shown considerable potential to close gaps in the HIV care cascade and community-based distribution of self-tests can enable programs to "reach people where they are." However, to maximize the effectiveness of HIVST, it will be necessary for self-tests to be used by PLHIV who are unaware of their status and persons at high risk of HIV infection. It will also be essential for those who obtain positive results to seek confirmatory testing and follow-up care. Interventions informed by behavioral economics can be easily incorporated into existing HIVST programs and warrant further testing as part of efforts to improve the effectiveness of HIVST.

## Supporting information

**S1 Checklist. CONSORT 2010 checklist of information to include when reporting a randomised trial**\*.
(DOC)

**S1 Text. Study protocol.**
(PDF)

## Acknowledgments

The authors gratefully acknowledge the Ezintsha STAR team for field and data support.

## Author Contributions

**Conceptualization:** Noora Marcus, Mothepane Phatsoane, Vanessa Msolomba, Naleni Rhagnath, Mohammed Majam, François Venter, Harsha Thirumurthy.

**Data curation:** Alison M. Buttenheim, Laura Schmucker, Noora Marcus.

**Formal analysis:** Alison M. Buttenheim, Laura Schmucker, Harsha Thirumurthy.

**Funding acquisition:** Alison M. Buttenheim, Harsha Thirumurthy.

**Investigation:** Alison M. Buttenheim, Mohammed Majam, François Venter, Harsha Thirumurthy.

**Methodology:** Alison M. Buttenheim, Noora Marcus, Mothepane Phatsoane, Vanessa Msolomba, Mohammed Majam, François Venter.

**Project administration:** Laura Schmucker, Noora Marcus, Mothepane Phatsoane, Vanessa Msolomba, Naleni Rhagnath, Harsha Thirumurthy.

**Supervision:** Alison M. Buttenheim, Noora Marcus, Mohammed Majam, Harsha Thirumurthy.

**Writing – original draft:** Alison M. Buttenheim.

**Writing – review & editing:** Alison M. Buttenheim, Laura Schmucker, Noora Marcus, Mothepane Phatsoane, Vanessa Msolomba, Naleni Rhagnath, Mohammed Majam, François Venter, Harsha Thirumurthy.

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
