## [Decision Letter · Decision Letter 0]

24 Jun 2022

PGPH-D-22-00213

Planning and commitment prompts to encourage reporting of HIV self-test results: A cluster randomized pragmatic trial in Tshwane District, South Africa

Dear Dr. Buttenheim,

Thank you for submitting your manuscript to PLOS Global Public Health. After careful consideration, we feel that it has merit but does not fully meet PLOS Global Public Health’s publication criteria as it currently stands. Therefore, we invite you to submit a revised version of the manuscript that addresses the points raised during the review process.

We look forward to receiving your revised manuscript.

Kind regards,

Leonardo Martinez

Academic Editor

Journal Requirements:

State the initials, alongside each funding source, of each author to receive each grant.

2. In the online submission form, you indicated that your data will be submitted to a repository upon acceptance.  We strongly recommend all authors deposit their data before acceptance, as the process can be lengthy and hold up publication timelines. Please note that, though access restrictions are acceptable now, your entire data will need to be made freely accessible if your manuscript is accepted for publication. This policy applies to all data except where public deposition would breach compliance with the protocol approved by your research ethics board. If you are unable to adhere to our open data policy, please kindly revise your statement to explain your reasoning and we will seek the editor's input on an exemption. Please be assured that, once you have provided your new statement, the assessment of your exemption will not hold up the peer review process.

3. Please provide separate figure files in .tif or .eps format and ensure that all files are under our size limit of 10MB.

4. We have noticed that you have uploaded Supporting Information files, but you have not included a list of legends. Please add a full list of legends for your Supporting Information files after the references list.

Additional Editor Comments (if provided):

Reviewer #1: This study which investigated an important question relating to action following self-testing for HIV is very interesting. While the research question is very relevant, I found a couple of pieces not speaking to the correct design.

1. The study is given as a cluster randomized controlled trial yet the sample size calculation and the analysis is done at individual level.

2. The manuscript does not follow any reporting guidelines such as the CONSORT Extension for cluster randomized trials.

3. The results section is simply too brief.

4. The outcome measurement is not properly given.

5. The study population is not completely described i.e. the inclusion and exclusion criteria is not well described.

6. The comparator is not very clear: is it the enhanced SOC or just SOC and how the implementation was done may have biased results because the authors say that they did intervention first and SOC later on each day.

Reviewer #2: A well written manuscript which addresses an important topic of results reporting around HIVST. It has been well established that HIVST does work in finding hard to reach populations, but linkage and results reporting remains a problem as rightly highlighted by the author. Even though there was a rise in the results reported in the intervention arms as compared to the standard arm, it would have been great if the author gave us more information on the characteristics of the participants. I suggest an addition of 2 tables, 1 for characteristics of those who took kits and another for those who reported results. In this way, a comparison can be made to see if there is a difference between those who did and those who did not report results which is important in the context of who we should expect to respond to these interventions.
---

## [Editor Report · Decision Letter 1]

23 Sep 2022

Planning and commitment prompts to encourage reporting of HIV self-test results: A cluster randomized pragmatic trial in Tshwane District, South Africa

PGPH-D-22-00213R1

Dear Dr. Buttenheim,

We are pleased to inform you that your manuscript 'Planning and commitment prompts to encourage reporting of HIV self-test results: A cluster randomized pragmatic trial in Tshwane District, South Africa' has been provisionally accepted for publication in PLOS Global Public Health.

Best regards,

Leonardo Martinez

Academic Editor